# Cell-autonomous targeting of arabinogalactan by host immune factors inhibits mycobacterial growth

**Lianhua Qin[1†], Junfang Xu[1,2†], Jianxia Chen[1,2†], Sen Wang[3†], Ruijuan Zheng[1], Zhenling Cui[1], Zhonghua Liu[1], Xiangyang Wu[1], Jie Wang[1], Xiaochen Huang[1], Zhaohui Wang[4], Mingqiao Wang[4], Rong Pan[4], Stefan HE Kaufmann[5,6,7], Xun Meng[4,8]\*, Lu Zhang[9]\*, Wei Sha[1,10]\*, Haipeng Liu[1,11]\***

[1]Shanghai Key Laboratory of Tuberculosis, Shanghai Pulmonary Hospital, Tongji University School of Medicine, Shanghai, China; [2]Clinical and Translational Research Center, Shanghai Pulmonary Hospital, Tongji University School of Medicine, Shanghai, China; [3]Department of Infectious Diseases, National Medical Centre for Infectious Diseases, National Clinical Research Centre for Aging and Medicine, Shanghai Key Laboratory of Infectious Diseases and Biosafety Emergency Response, Huashan Hospital, Fudan University, Shanghai, China; [4]Abmart Inc, Shanghai, China; [5]Max Planck Institute for Infection Biology, Berlin, Germany; [6]Max Planck Institute for Multidisciplinary Sciences, Göttingen, Germany; [7]Hagler Institute for Advanced Study, Texas A&M University, College Station, United States; [8]Multitude Therapeutics, Shanghai, China; [9]School of Life Science, Fudan University, Shanghai, China; [10]Department of Tuberculosis, Shanghai Pulmonary Hospital, Tongji University School of Medicine, Shanghai, China; [11]Central Laboratory, Shanghai Pulmonary Hospital, Tongji University School of Medicine, Shanghai, China

**\*For correspondence:**
xun.meng@ab-mart.com (XM);
zhanglu407@fudan.edu.cn (LZ);
shfksw@126.com (WS);
haipengliu@tongji.edu.cn (HL)

†These authors contributed equally to this work

## eLife Assessment

The main idea tested in this work is that host galectin-9 inhibits Mycobacterium tuberculosis (Mtb) growth by recognizing the Mtb cell wall component arabinogalactan (AG) and, as a result, disrupting mycobacterial cell wall structure. Moreover, a similar effect is achieved by anti-AG antibodies. While the hypothesis is intriguing and the work has the potential to make a **valuable** contribution to Mtb therapy, the evidence presented is **incomplete** and does not explain several critical points including the dose-independent effect of galectin-9 on Mtb growth and how anti-AG antibodies and galectin-9 access the AG layer of intact Mtb.

**Abstract** Deeper understanding of the crosstalk between host cells and *Mycobacterium tuberculosis* (Mtb) provides crucial guidelines for the rational design of novel intervention strategies against tuberculosis (TB). Mycobacteria possess a unique complex cell wall with arabinogalactan (AG) as a critical component. AG has been identified as a virulence factor of Mtb which is recognized by host galectin-9. Here, we demonstrate that galectin-9 directly inhibited mycobacterial growth through AG-binding property of carbohydrate-recognition domain 2. Furthermore, IgG antibodies with AG specificity were detected in the serum of TB patients. Based on the interaction between galectin-9 and AG, we developed a monoclonal antibody (mAb) screening assay and identified AG-specific mAbs which profoundly inhibit Mtb growth. Mechanistically, proteomic profiling and morphological characterizations revealed that AG-specific mAbs regulate AG biosynthesis, thereby inducing cell wall swelling. Thus, direct AG-binding by galectin-9 or antibodies contributes to protection against

TB. Our findings pave the way for the rational design of novel immunotherapeutic strategies for TB control.

## Introduction

TB, caused by Mtb remains a considerable threat to human health. In 2021, 10.6 million people fell ill with TB, 1.6 million people died from the disease, and 450,000 new TB cases suffered from rifampicin-resistant or multidrug-resistant TB on this globe (*Bagcchi, 2023*). Although TB treatment measures are in place in many parts of the world, cure rates are insufficient due to the increasing emergence of drug resistance. Therefore, novel intervention strategies are urgently needed. Rational design of novel TB therapeutics depends on a better understanding of the crosstalk between Mtb and host cells.

Following inhalation of aerosols carrying Mtb, innate immune responses are initiated which constitute a first line of defense. First, Mtb are engulfed by mononuclear phagocytes into the phagosome which matures and then fuses with lysosomes (*Chandra et al., 2022*). The pathogen-associated molecular patterns (PAMPs) from Mtb are recognized by a variety of pattern recognition receptors (PRRs), such as Toll-like receptors (TLRs), Nod-like receptors (NLRs), C-type lectins receptors (CLRs) and cyclic GMP-AMP synthase (cGAS), resulting in production of inflammatory cytokines, chemokines and anti-bacterial peptides to restrict bacterial growth (*Reiling et al., 2002*; *Fremond et al., 2004*; *Wilson et al., 2015*; *Watson et al., 2015*). Mtb can escape from the phagosome into the cytosol and can be recaptured in autophagosomes, through a process termed xenophagy, to form a degradative autolysosome (*Wang and Li, 2020*; *Lopez et al., 2018*). Cell-autonomous defense mechanisms also include the production of reactive oxygen and nitrogen intermediates, hypoxia, mild acidity, and nutrition deprivation (*Nathan and Shiloh, 2000*; *Lupoli et al., 2018*). These mechanisms help to limit the growth and spread of the bacteria within the cell, and contribute to the initiation of an adaptive immune response. However, our knowledge about host immune factors that target Mtb components and directly inhibit replication is limited.

Mtb can evade and resist immune defense by entering a state of dormancy which can last for years. This is due to the capacity of Mtb to synthesize a sturdy cell wall, slow down metabolism to promote growth arrest, and implement the so-called stringent response (*Batt et al., 2020*; *Hauryliuk et al., 2015*). These mechanisms provide the basis for the long-term persistence of Mtb until immune control weakens. Once immune control deteriorates, Mtb acquires a metabolic active stage and induces progression to active TB.

The complex cell wall of Mtb provides a barrier not only for host defense, but also for antibiotics. Accordingly, components of the cell wall are well-established drug targets. The essential core cell wall structure is composed of three distinct layers: (a) the cross-linked network of peptidoglycan (PGN), (b) the highly branched AG polysaccharide, and (c) the characteristic long-chain mycolic acids (*Jankute et al., 2015*). Among these, AG has been an important target for anti-TB drugs, though understanding of its biological functions is limited. E.g., ethambutol, one of the front-line anti-TB drugs, targets the arabinosyltransferases EmbA, EmbB, and EmbC, which are critical for AG synthesis (*Escuyer et al., 2001*; *Goude et al., 2009*; *Zhang et al., 2020*). However, it remains unclear whether AG is directly targeted by natural host immune factors in TB.

Recently, we identified AG as a virulence factor of Mtb that is recognized by galectin-9, a member of the β-galactoside binding gene family. Upon AG binding, galectin-9 initiates the downstream TAK1-ERK-MMP signaling cascade leading to pathologic impairment of the lung (*Wu et al., 2021*). Galectin-4 and galectin-8 directly kill *E. coli* by recognizing blood group antigens of bacteria (*Stowell et al., 2010*). This raises the question of whether galectin-9 inhibits mycobacterial growth via targeting AG. Here, we demonstrate a novel cell-autonomous mechanism by which galectin-9 impedes mycobacterial growth via its AG-binding property in a carbohydrate recognition domain (CRD) 2-dependent mode. Moreover, in sera of TB patients, we identified anti-AG IgG antibodies which were supposed to defend against TB. Employing a monoclonal antibody (mAb) screening array, we identified anti-AG mAbs, CL010746 and CL046999, which were capable of restraining Mtb growth by regulating AG biosynthesis. Thus, AG possesses characteristic features of protective antigens. In sum, our work identified a previously unknown role of galectin-9 and anti-AG antibodies in TB control. Hence, our findings provide the basis for the rational design of mAb-based immunotherapy of TB as a novel approach toward host-directed therapy of one of the deadliest infectious diseases globally.

# Results

## Galectin-9 inhibits mycobacterial growth

Our previous work demonstrated that galectin-9 directly interacts with AG and AG-containing bacteria (*Wu et al., 2021*). Given the capacity of galectin-4 and galectin-8 to kill *E. coli* (*Stowell et al., 2010*), we interrogated whether galectin-9 directly interferes with mycobacterial replication. Real-time monitoring of in vitro cultures revealed that recombinant galectin-9 protein inhibits the growth of Mtb at a concentration as low as 10 ng/mL (*Figure 1A*). Native structural conformation of galectin-9 was required for its bacteriostatic effect since heat inactivation at 95°C for 5 min abrogated this activity (*Figure 1B*). CFU assays further validated that galectin-9 inhibited Mtb growth (*Figure 1C*). Galectin-9 also impaired the replication of the fast-growing *Mycobacterium smegmatis* in a dose-dependent manner (*Figure 1D*). Consistently, ELISA revealed a profoundly higher abundance of galectin-9 in serum from TB patients than that from heathy donors, implying that galectin-9 contributes to resistance against Mtb infection (*Figure 1E*). Of note, the average concentration of galectin-9 in the sera of healthy donors was 3.602 ng/mL (*Figure 1E*). We speculate that this high abundance of galectin-9 contributes to the maintenance of latent TB infection by restricting Mtb spreading from granuloma, where the pathogen is contained. In sum, we conclude that galectin-9 directly inhibits mycobacterial growth.

Macrophages are part of the first line of defense against invading mycobacteria. When human monocytic THP-1 cells were infected with *Mycobacterium bovis* BCG fused with DsRed, immunofluorescence assays revealed recruitment of galectin-9 to mycobacteria in a time-dependent manner (*Figure 1F and G*). In line with this observation, a robust accumulation of galectin-9 around invading Mtb H37Rv-GFP was also observed in THP-1 cells post infection (*Figure 1H and I*). Though galectin-9 has been reported to be critical for initiation of mTOR signaling and induction of autophagy (*Jia et al., 2018*; *Jia et al., 2020*; *Bell et al., 2021*), our work revealed a novel cell-autonomous mechanism whereby galectin-9 recruitment restricts mycobacterial growth in an autophagy-independent manner.

## Carbohydrate recognition is essential for galectin-9-mediated inhibition of mycobacterial growth

Given that galectin-9 binds to β-galactoside, we interrogated whether carbohydrate recognition by galectin-9 is essential for inhibition of mycobacterial growth. Addition of lactose rich in β-galactoside (generally used for neutralization of carbohydrate-binding of galectin-9) completely reversed mycobacterial growth inhibition by galectin-9 in vitro (*Figure 2A*). In contrast, the addition of glucose had no such effect (*Figure 2B*). These results indicate that the β-galactose binding property of galectin-9 is involved in mycobacterial activity. Moreover, the addition of AG reversed growth inhibition by galectin-9 (*Figure 2C*), indicating that the AG-binding property of galectin-9 is involved in an anti-mycobacterial activity. Our previous work had demonstrated that CRD2, but not CRD1, of galectin-9, mediated its interaction with AG (*Wu et al., 2021*). As expected, the addition of purified CRD2, but not of CRD1, to some extent hindered Mtb growth, emphasizing that AG binding to galectin-9 was sufficient for anti-mycobacterial effects (*Figure 2D*). Taken together, carbohydrate recognition is essential for galectin-9-mediated inhibition of mycobacterial growth.

## Identification of anti-AG antibodies from TB patients

Given the higher abundance of galectin-9 in serum from active TB patients, we next interrogated whether anti-AG antibodies are present in the serum of TB patients. An ELISA assay was developed for the identification of anti-AG antibodies by coating AG on plates (*Figure 3A*). An adequate window of serum dilutions allowed a linear correlation between $OD_{450}$ and dilution over an appropriate range (*Figure 3B*). Subsequently, we determined the relative abundance of anti-AG serum antibodies in 17 healthy donors and 25 active TB patients, all of whom had received BCG vaccination within 24 hr after birth. Our findings revealed a significant increase in the abundance of anti-AG antibodies among TB patients when compared to healthy donors (*Figure 3C*). We speculate that during Mtb infection, anti-AG IgG antibodies are induced, which potentially contribute to protection against TB by directly inhibiting Mtb replication albeit seemingly in vain. Whether anti-AG antibody levels at the site of Mtb growth in patients are too low or whether growth inhibition is nullified by excessive replication of Mtb remains to be solved in the future.

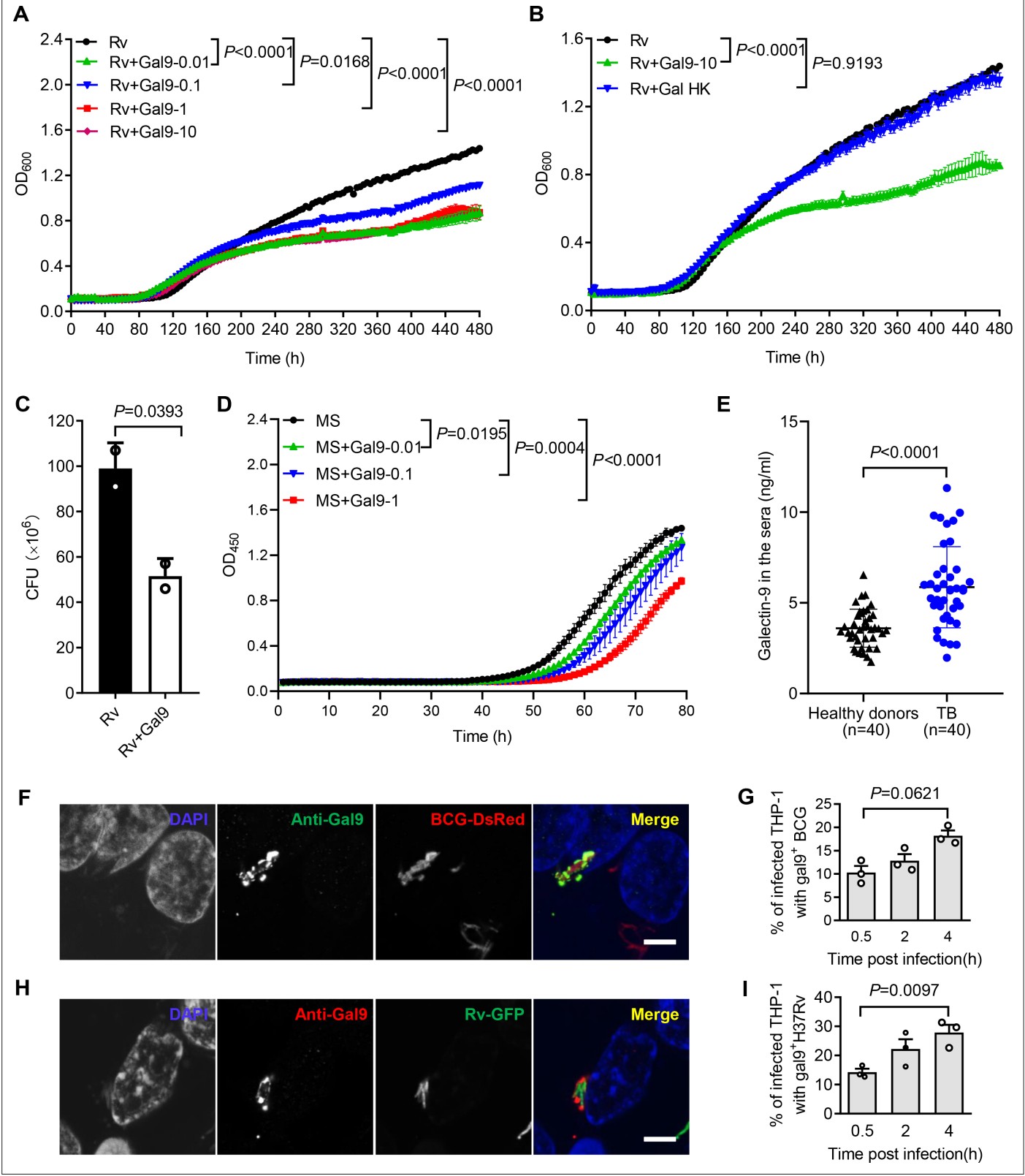

**Figure 1.** Galectin-9 inhibits mycobacterial growth directly. (**A**) Profile of *Mycobacterium tuberculosis* (Mtb) H37Rv (Rv) grown at 37 °C in Middlebrook 7H9 liquid medium with different concentrations of Galectin-9 (Gal9, 0, 0.01, 0.1, 1, 10 μg/mL). Growth curve was measured using a Bioscreen Growth Curve Instrument. Optical density was measured at absorbance at 600 nm every 2 hr. (**B**) Growth profile of Mtb H37Rv (Rv) in Middlebrook 7H9 liquid medium with10 μg/mL galectin-9 (Gal9) or inactivated galectin-9 (Gal9 HK, heat-killed at 95℃ for 5 min). (**C**) CFU of Mtb H37Rv (Rv) on Middlebrook

*Figure 1 continued on next page*

*Figure 1 continued*

7H10 solid medium after being incubated in Middlebrook 7H9 liquid medium with or without 10 µg/mL Galectin-9 (Gal9) for 30 hr at 37°C. Cultures were grown at 37 °C for 4 weeks for enumeration of CFU. (**D**) Growth profile of *Mycobacterium smegmatis* (MS) in Middlebrook 7H9 liquid medium with different concentrations of Galectin-9 (Gal9, 0, 0.01, 0.1, 1 µg/mL). (**E**) Concentrations of galectin-9 in sera of healthy donors (n=40) and active TB patients (n=40). (**F**) Confocal microscopy of *M. bovis* BCG-DsRed (BCG-DsRed, red) and Galectin-9 (Anti-Gal9, green) in THP-1 cells. Nuclei was stained with DAPI (blue). (**G**) Percent of cells with galectin-9 positive (gal9+) BCG in total infected THP-1 cells. Symbols indicate a colocalization ratio of at least 12 fields in each experiment. (**A, H**) Confocal microscopy of Mtb H37Rv-GFP (Rv-GFP, green) and Galectin-9 (Anti-Gal9, red) in THP-1 cells. Nuclei were stained with DAPI (blue). 63x magnification. Scar bar, 5µm. (**I**) Percent of cells with galectin9 positive (gal9+) Mtb H37Rv in total infected THP-1 cells. Symbols indicate a colocalization ratio of at least 12 fields in each experiment. Data are shown as mean ± SD, n=3 biologically independent experiments performed in triplicate (**A–D**). Data are representative of three independent experiments with similar results (**F and H**). Two-tailed unpaired Student's t-test (A-D, G, and I) or Mann-Whitney U test (**E**). p<0.05 was considered statistically significant.

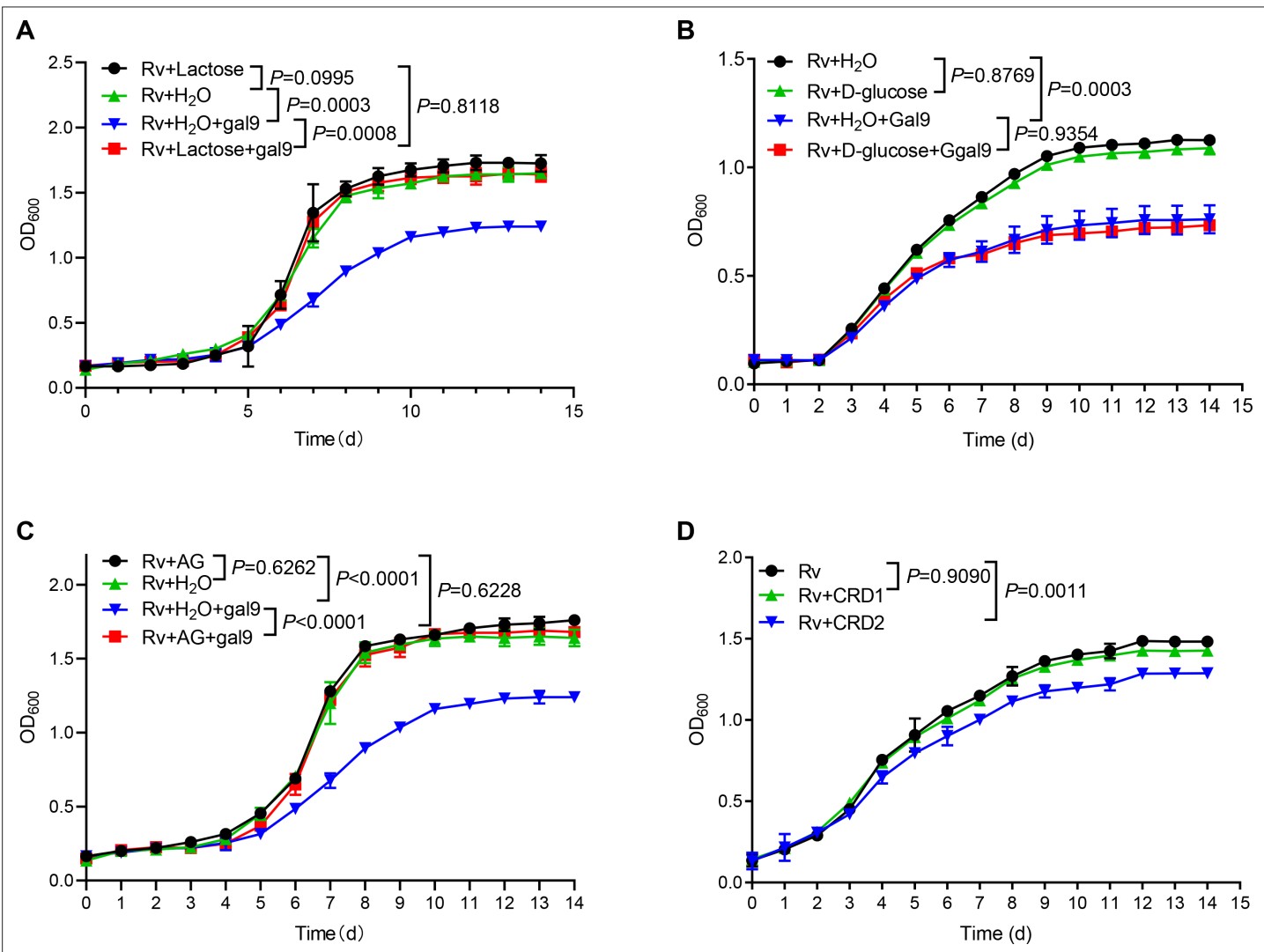

**Figure 2.** Carbohydrate recognition is essential for galectin-9-mediated inhibition of *Mycobacterium tuberculosis* (Mtb) growth. (**A**) Growth profile of Mtb H37Rv (Rv) in Middlebrook 7H9 liquid medium with or without galectin-9 (Gal9, 10 µg/mL) and lactose (1 µg/mL). (**B**) Growth profile of Mtb H37Rv (Rv) in Middlebrook 7H9 liquid medium with or without galectin-9 (Gal9, 10 µg/mL) and D-glucose (10 µg/mL). (**C**) Growth profile of Mtb H37Rv (Rv) in Middlebrook 7H9 liquid medium with or without galectin-9 (Gal9, 10 µg/mL) and AG (1 µg/mL). (**D**) Growth profile of Mtb H37Rv (Rv) in Middlebrook 7H9 liquid medium with 1 µg/mL CRD1 or CRD2 of galectin-9. Data are shown as mean ± SD, n=3 biologically independent experiments performed in triplicate (**A–D**). Two-tailed unpaired Student's t test (**A–D**). p<0.05 was considered statistically significant.

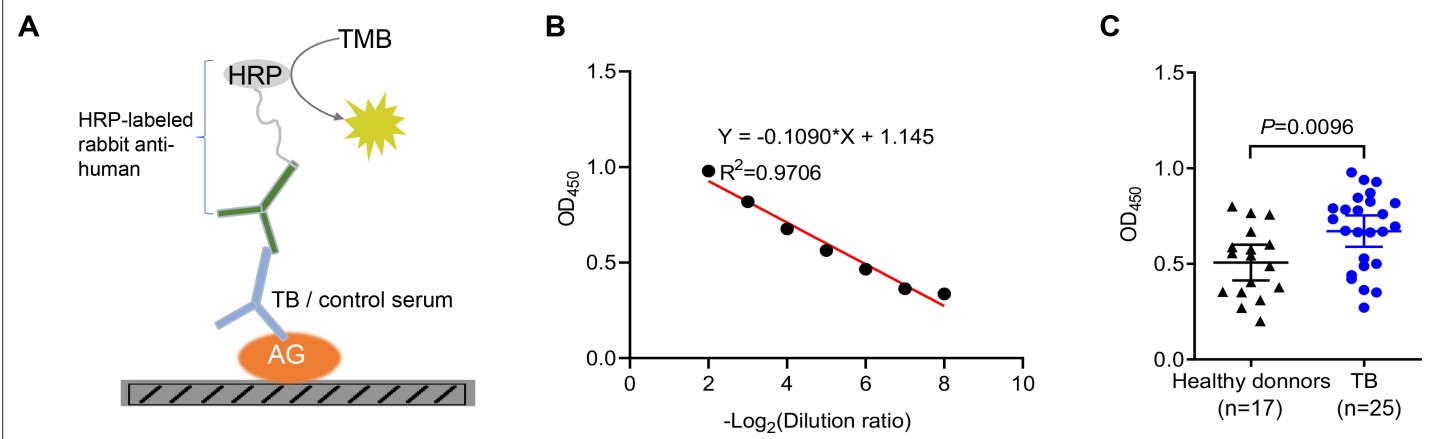

**Figure 3.** Identification of anti-arabinogalactan (AG) antibodies from tuberculosis (TB) patients. (**A**) Schematic presentation of ELISA assay for detecting anti-AG IgG antibodies in the serum of TB patients. (**B**) Linear correlation between OD and serum dilution ratio determined by ELISA assay. (**C**) Anti-AG IgG antibody levels in TB patients (n=25) and healthy BCG-immunized controls (n=17) were determined via ELISA. Data are representative of three independent experiments with similar results (**B**). Mann-Whitney U test (**C**). p<0.05 was considered statistically significant.

## Generation of anti-AG mAb

Based on the finding that AG binding of galectin-9 inhibits mycobacterial growth, we embarked on the development of anti-AG mAbs with blocking activity. Given the high affinity between galectin-9 and AG, we developed an antibody chip comprising 62,208 mAbs to screen for anti-AG activity (***Wu et al., 2021***). Briefly, the antibody chip was incubated with AG, and bound AG was subsequently detected using galectin-9 in conjunction with FITC-labeled anti-galectin-9 monoclonal antibody (***Figure 4A***). We filtered out 12 candidate mAbs exhibiting binding affinity to AG (***Figure 4A and B***). Subsequently, we validated their AG-binding capacity using ELISA. AG was coated on plates, and mAbs were added in twofold serial dilutions (***Figure 4C***). The ELISA assay revealed a robust AG-binding curve for CL010746 (referred to as mAb1) and CL046999 (referred to as mAb2) (***Figure 4D***). Furthermore, both mAbs exhibited specific binding to Mtb H37Rv-GFP as demonstrated by immunofluorescent assay (***Figure 4E and F***). Therefore, we have successfully developed anti-AG mAbs that bind Mtb directly.

## Anti-AG antibody inhibits Mtb growth

For functional characterization, we monitored the in vitro mycobacterial growth in the presence or absence of anti-AG mAbs. Both mAb1 and mAb2 demonstrated inhibition of Mtb growth (***Figure 5A***). This finding was further confirmed through CFU determination (***Figure 5B***). Likewise, both mAbs markedly inhibited the growth of *Mycobacterium smegmatis* as evidenced by real-time OD monitoring (***Figure 5C***) and CFU assay (***Figure 5D***). In conclusion, the newly identified anti-AG mAbs demonstrated direct blockade of mycobacterial growth through binding to AG. Direct inhibitory effects on Mtb growth by anti-AG antibodies emphasize that AG expresses features of protective antigens.

## Proteomics profiling of the mycobacterial response to anti-AG mAb

To elucidate the molecular mechanisms underlying the inhibition of mycobacterial growth by anti-AG antibodies, we conducted proteomic profiling of Mtb in response to anti-AG mAb1 treatment. Gene ontology (GO) enrichment analysis revealed significant enrichment of numerous cellular and metabolic processes, primarily related to the biosynthesis of the outer membrane, upon treatment with anti-AG mAb1. These processes include cell periphery, external encapsulating structure, organic substance metabolic process, cellular metabolic process, primary metabolic process, nitrogen compound metabolic process, and biosynthetic process (***Figure 6A***). Moreover, the formamidopyrimidine-DNA glycosylase N-terminal domain was enriched based on the analysis of functional enrichment and protein domain of differentially expressed antigens (***Figure 6B and C***). Additionally, the KEGG pathway analysis demonstrated a significant enrichment of lipoarabinomannan (LAM) biosynthesis pathways (***Figure 6D***). Consistently, the upregulated proteins Rv0236.1 and Rv3806c were involved in the biosynthesis of the mycobacterial cell wall arabinan (***Figure 6E***). In the re-annotated

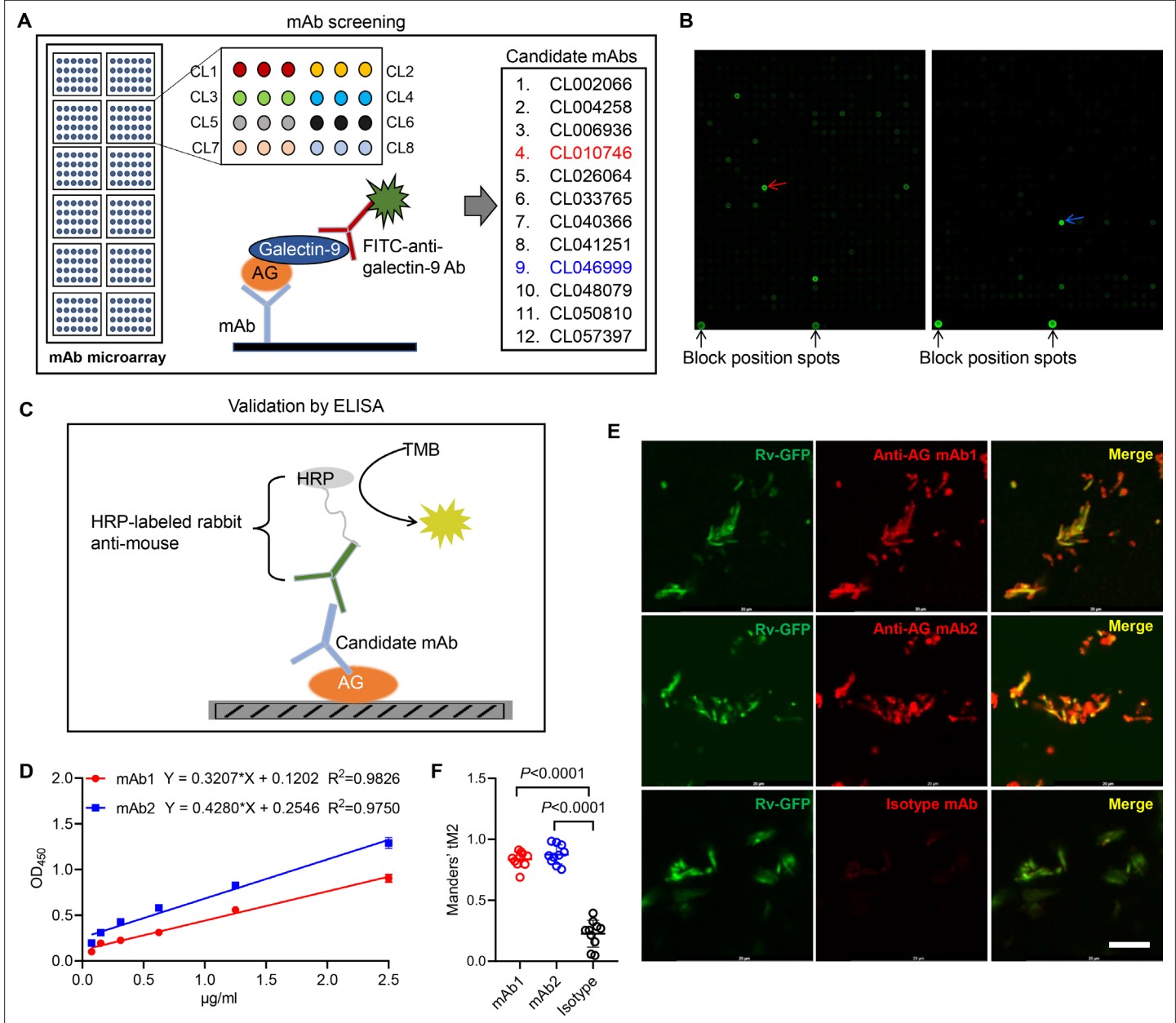

**Figure 4.** Development of anti-arabinogalactan (AG) mAbs. (**A**) Schematic presentation of mAb screening for AG specificity. (**B**) Representative image of chip hybridization for mAb screening. Bright spots in the bottom mark the end line of each array block. Other spots represent AG binding to mAbs. CL010746 (mAb1) and CL046999 (mAb2) were labeled with red arrow and blue arrow, respectively. (**C**) Schematic presentation of candidate anti-AG mAbs validation by ELISA. (**D**) Binding curve of mAb1 and mAb2 to AG was determined by ELISA assay. (**E**) Confocal microscopy of *Mycobacterium tuberculosis* (Mtb) H37Rv-GFP (Rv-GFP, green) and anti-AG mAbs (red). 100x oil immersion.Scar bar, 10 µm. (**F**) Quantification of colocalization between anti-AG mAb and Mtb H37Rv-GFP by calculating Mander's coefficients in (**E**). tM2, Mander's coefficient of red above the autothreshold of green. Data are representative of three independent experiments with similar results (**D, E**). Data are shown as mean ± SD, n=10 (**F**). Two-tailed unpaired Student's t-test (**F**). p<0.05 was considered statistically significant.

genome sequence of Mtb, Rv0236.1 consists of Rv0236c and Rv0236A. Rv0236c is predicted to be a cognate of the GT-C superfamily of glycosyltransferases and likely acts as arabinofuranosyltransferase involved in AG synthesis (*Skovierová et al., 2009*). On the other hand, Rv0236A is a small secreted protein involved in cell walls and cell processes (*Marmiesse et al., 2004*). Additionally, Rv3806c is a decaprenylphosphoryl-5-phosphoribose (DPPR) synthase involved in AG synthesis (*He et al., 2015*). These data provide compelling evidence to suggest that anti-AG mAbs regulate AG biosynthesis.

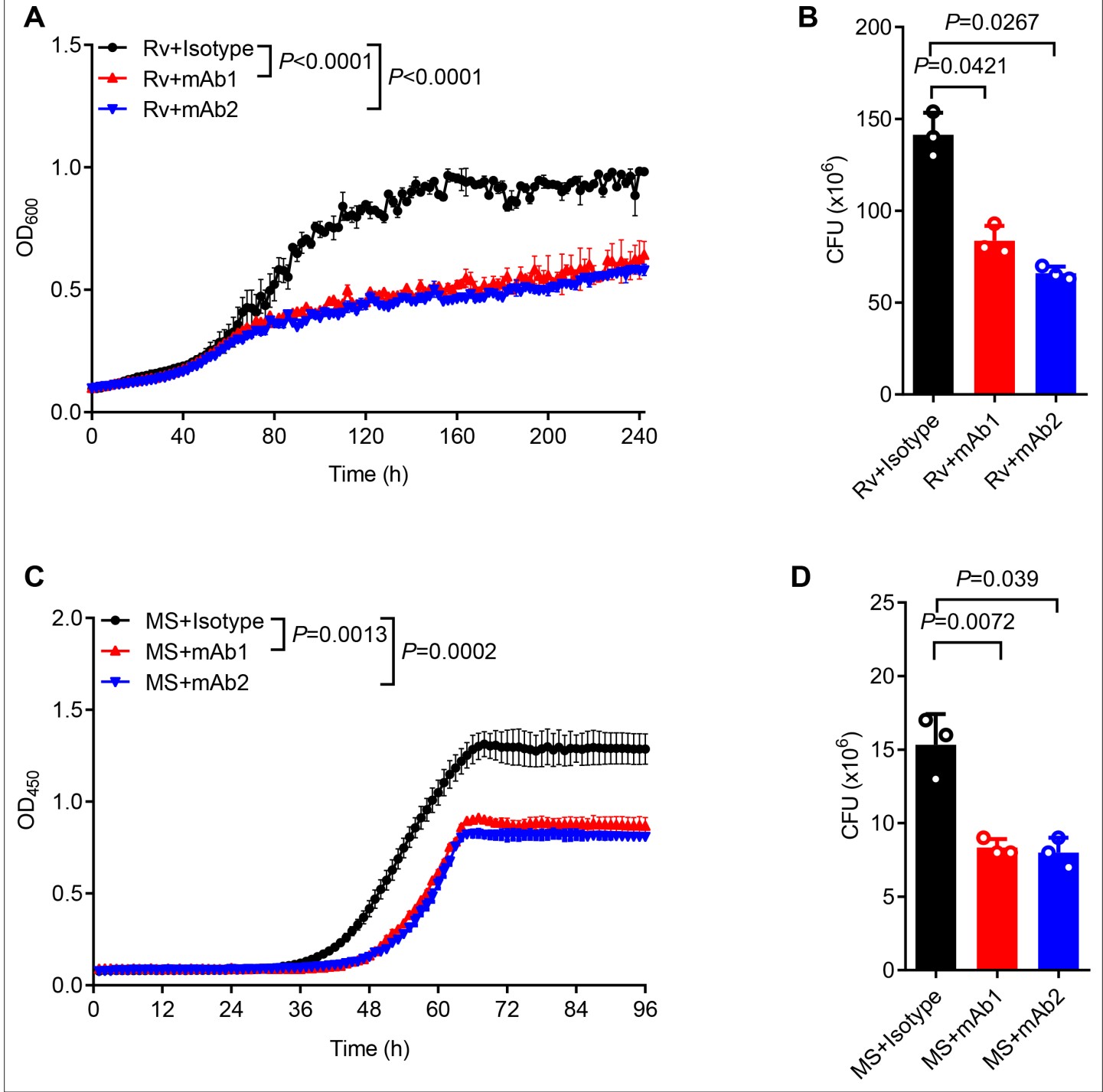

**Figure 5.** Anti-arabinogalactan (AG) antibody inhibits mycobacterial growth. (**A**) Growth profile of *Mycobacterium tuberculosis* (Mtb) H37Rv (Rv) in Middlebrook 7H9 liquid medium with or without mAb1/mAb2 (1 μg/mL). (**B**) CFU of Mtb H37Rv (Rv) on Middlebrook 7H10 solid medium with or without mAb1/mAb2 (1 μg/mL). Cultures were grown at 37 °C for 4–8 weeks. (**C**) Growth profile of *Mycobacterium smegmatis* (MS) in Middlebrook 7H9 liquid medium with or without mAb1/mAb2 (1 μg/mL). (**D**) CFU of *Mycobacterium smegmatis* (MS) on Middlebrook 7H10 solid medium with or without mAb1/mAb2 (1 μg/mL). Cultures were grown at 37 °C for 5–10 days. Data are shown as mean ± SD, n=3 (**A, C**) and n=3 biologically independent experiments performed in triplicate (**B, D**). Two-tailed unpaired Student's t-test (**A–D**). p<0.05 was considered statistically significant.

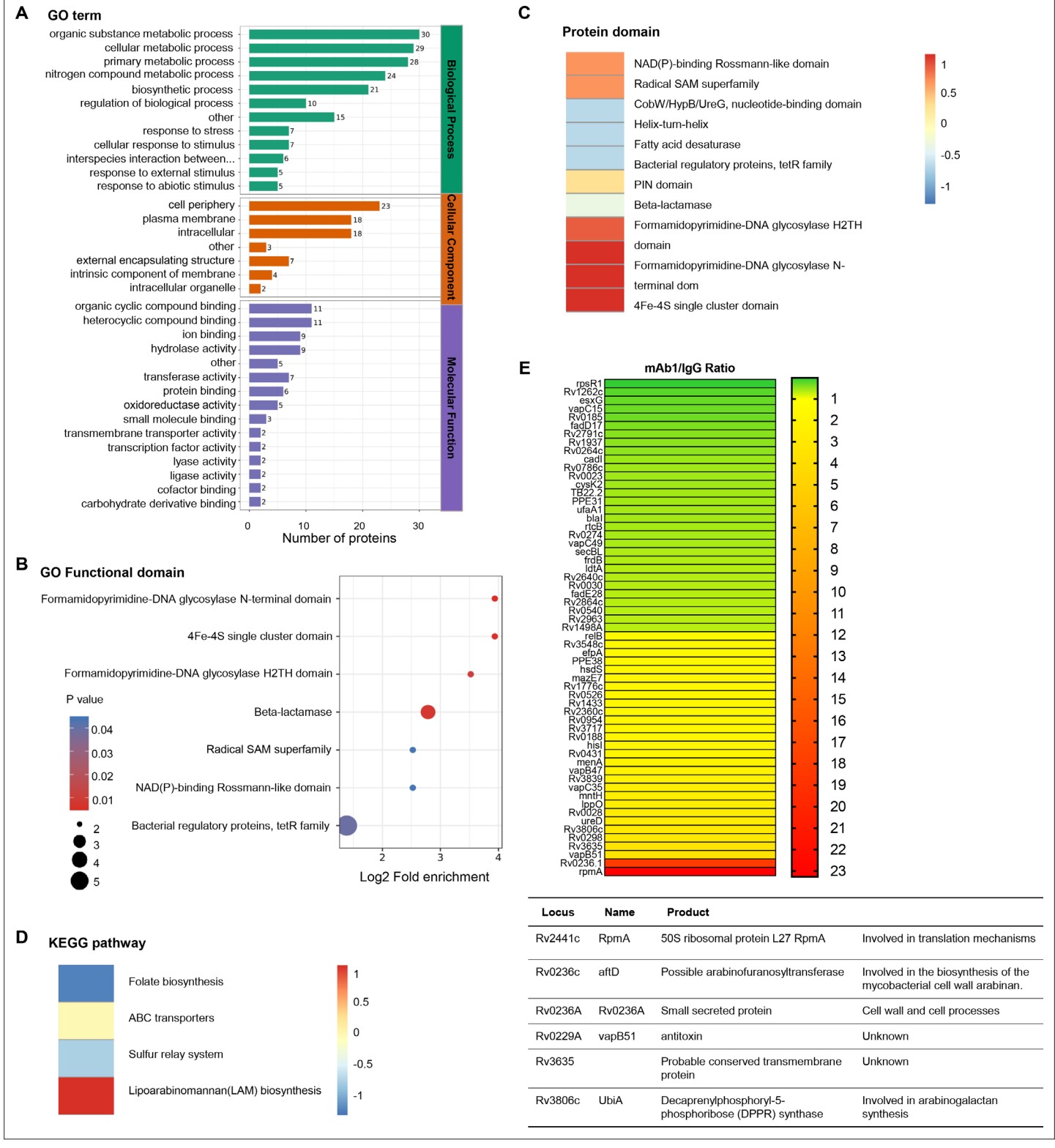

**Figure 6.** Proteomics profiling of the response of *Mycobacterium tuberculosis* (Mtb) to anti-arabinogalactan (AG) antibody. (**A**) Gene ontology (GO) class of differentially expressed proteins in Mtb H37Rv treated with mAb1 (1 μg/mL) for 30 hr followed by proteomics analysis. IgG was set as control. (**B**) Functional enrichment of differentially expressed proteins in Mtb H37Rv in (**A**). (**C**) Protein domain of differentially expressed proteins in Mtb H37Rv in (**A**). (**D**) KEGG class of differentially expressed proteins in Mtb H37Rv in (**A**). (**E**) Upregulation or downregulation genes in Mtb H37Rv in (**A**).

## Targeting AG by mAbs modulates the cell wall of Mtb

To verify the impact of anti-AG mAbs on the biosynthesis of the mycobacterial cell wall, we characterized the morphological changes of Mtb treated with or without mAbs. Intriguingly, mAb1 and mAb2 treatment both led to a dispersed distribution of Mtb in cultures (*Figure 7A*). Acid-fast staining further revealed the formation of a cord-like structure in Mtb treated with mAb1 or mAb2, which was not observed following ethambutol (EMB) treatment (*Figure 7B*). Moreover, electron microscopy demonstrated that anti-AG mAbs treatment markedly increased the thickness of the Mtb cell wall (*Figure 7C and D*). Based on these findings, we conclude that targeting AG by specific antibodies, and likely by galectin-9 as well, impairs the growth of Mtb and other mycobacteria by modulating cell wall structure.

## Discussion

Mtb, the etiologic agent of TB, is one of the leading causes of death worldwide, further aggravated by increasing incidences of antibiotic resistance (*Singh and Chibale, 2021*; *Miotto et al., 2018*). Hence, TB remains a major contributor to the global disease burden. Host-directed therapy is increasingly recognized as an alternative or adjunct to antibiotic therapy (*Kaufmann et al., 2018*). Therefore, deeper insights into the interactions between Mtb and the host immune system are warranted. We previously demonstrated that mycobacterial AG binds to the galactoside-binding protein galectin-9, causing pathologic impairments in the lung via the TAK1-ERK-MMP signaling pathway (*Wu et al., 2021*). Here, we demonstrate that galectin-9 directly impedes mycobacterial growth through its AG-binding property. Furthermore, we identified natural anti-AG antibodies in sera of TB patients, which are predicted to inhibit Mtb growth. Based on these findings, we generated mAbs capable of binding AG and hindering Mtb replication. Proteomics profiling of Mtb revealed that the binding of anti-AG antibodies regulates AG biosynthesis which leads to swelling of the cell wall, as validated by morphological characterization. We conclude that galectin-9 and anti-AG antibodies serve as immune factors that restrain bacterial growth by targeting AG in the cell wall. Increasing evidence suggests a role for antibodies in protection against TB (*Lu et al., 2019*; *Irvine et al., 2021*). It is generally assumed that the role of antibodies in TB is based on their interactions with macrophages, which promote anti-mycobacterial activities such as phagolysosome fusion and production of reactive oxygen and nitrogen intermediates (*Chandra et al., 2022*; *Nathan and Shiloh, 2000*). In striking contrast, our findings demonstrate that anti-AG antibodies directly impair Mtb growth and thus emphasize that AG comprises features of protective antigens.

Galectins are a highly conserved class of molecules that play critical roles in multiple biological processes. Fifteen different types of galectins are known in humans, which can be classified based on their structure, subcellular localization, and function. For instance, galectin1 participates in regulating cell proliferation, apoptosis, and immune responses through interactions with specific glycosylated receptors on the cell surface, such as integrins and CD45 (*Cedeno-Laurent et al., 2012*; *Perillo et al., 1995*; *Ge et al., 2016*). More recently, it was shown that galectin-4 disrupts bacterial membranes and kills *E. coli* through interactions with lipopolysaccharides on the bacterial outer membrane (*Stowell et al., 2010*). Here, we demonstrate that galectin-9 significantly inhibits the replication of Mtb by interacting with AG in Mtb via its CRD2 domain. Similar to galectin-9, galectin-4, galectin-6, and galectin-8 also comprise 2 CRDs in tandem connected by a linker sequence (*Leffler et al., 2002*). It remains to be explored whether and how these galectins exert anti-mycobacterial activities via the CRD2 domain, thereby providing general insights into the role of galectin family cognates in immunity to TB.

After phagocytosis by pulmonary macrophages of the newly infected host, Mtb ends up inside phagosomes, where it downregulates its metabolism and enters a non-replicating persistent (NRP) state, termed dormancy, in response to host stress (*Russell, 2001*; *Gengenbacher and Kaufmann, 2012*). Once the immune response 'breaks down,' Mtb transits into a metabolically active and replicative state which ultimately results in progression to active TB disease (*van der Wel et al., 2007*). We demonstrated that galectin-9 accumulates around invading mycobacteria in host cells. However, whether Mtb recruits galectin-9 during dormancy, its active stage, or during both stages, has not been investigated. Of note, AG is hidden by mycolic acids in the outer layer. We speculate that during Mtb

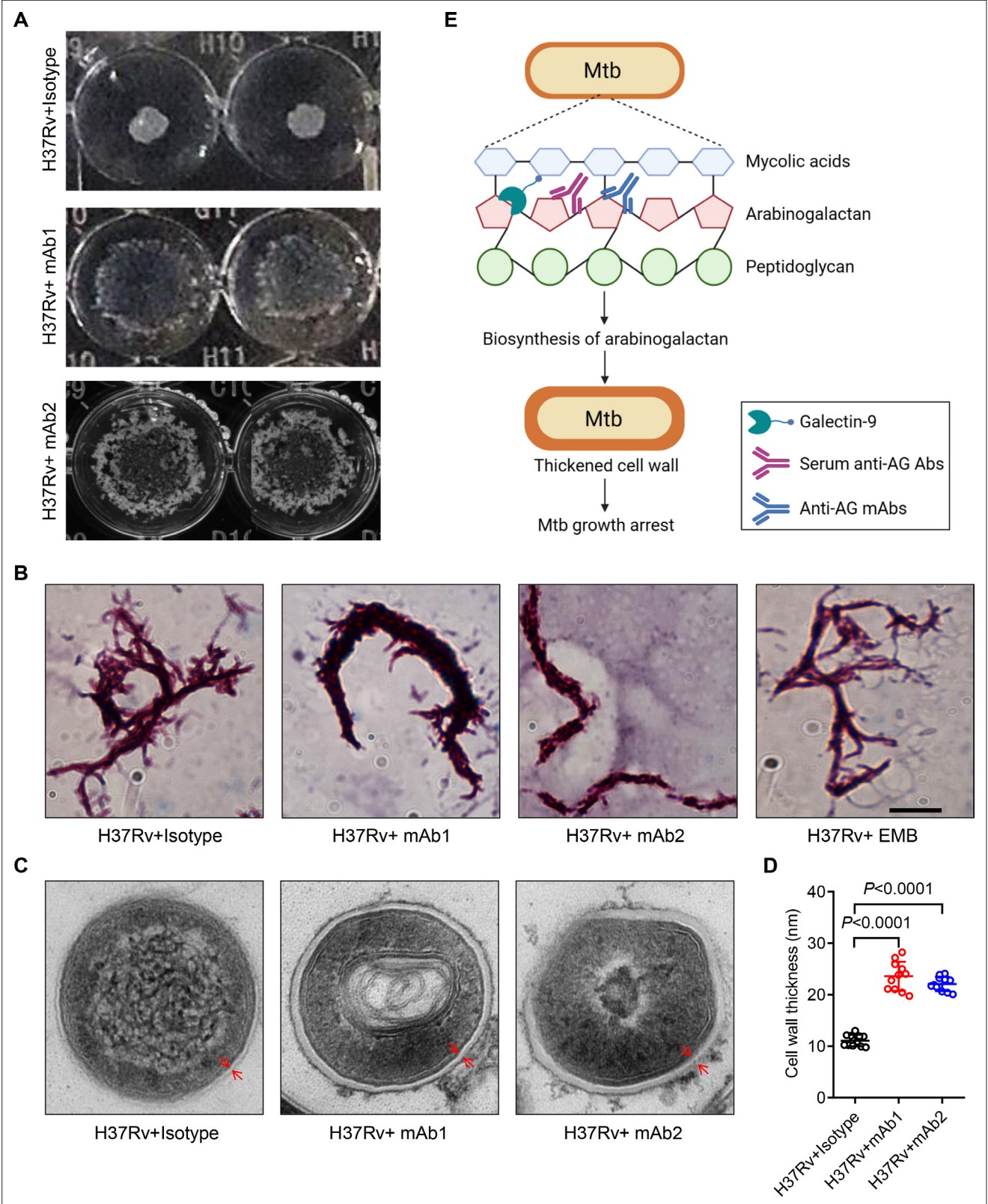

**Figure 7.** Mtb cell wall modulation by anti-arabinogalactan (AG) antibodies. (**A**) Morphologic characteristics for *Mycobacterium tuberculosis* (Mtb) H37Rv strain grown in liquid culture with or without anti-AG mAbs (1 µg/mL) observed by 2x magnifier. (**B**) Bacterial shape of Mtb H37Rv strain treated as in (**A**) was observed by acid-fast staining under a Leica DM2500 microscope using the 100x oil microscopy. EMB, Ethambutol. Scale bar, 20 µm. (**C**) Ultrastructural morphology of Mtb H37Rv treated as in (**A**) analyzed by transmission electron microscopy (TEM). The cell wall was labeled with

*Figure 7 continued on next page*

*Figure 7 continued*

red arrows. (**D**) Cell wall thickness of bacteria in (**C**). (**E**) Schematic presentation of Mtb growth arrest by Galectin-9 or anti-AG antibodies. Data are representative of three independent experiments with similar results (**A, B, and C**). Data are means ± SD of 11 bacteria, representatives of three independent experiments (**D**). Two-tailed unpaired Student's t-test (**D**). p<0.05 was considered statistically significant.

replication, cell wall synthesis is active and AG becomes exposed, thereby facilitating its binding to galectin-9 and leading to Mtb growth arrest.

Major drugs in clinical use for TB treatment inhibit Mtb growth by targeting different essential components and processes. For instance, Isoniazid inhibits mycolic acid synthesis by targeting InhA enzyme (*Quémard et al., 1995*), Ethambutol blocks AG biosynthesis by targeting EmbCAB complex *Telenti et al., 1997*, and Pyrazinamide disrupts the pH balance within the bacterial cell, thereby impairing mycobacterial growth (*Zhang et al., 1999*). Although the mycobacterial cell wall has been widely exploited as an antibiotic target, to date, drugs that directly bind AG and inhibit Mtb growth have not been reported. Here, we identified host galectin-9 and anti-AG antibodies (both serum antibodies from patients and mAbs) which recognize AG and thus inhibit Mtb replication. Hence, anti-AG mAbs can be harnessed for the design of novel biologics which address the challenge of drug resistance in TB.

We observed elevated levels of galectin-9 in the serum of active TB patients, consistent with reports indicating that cleaved galectin-9 levels in the serum serve as a biomarker for severe infection (*Iwasaki-Hozumi et al., 2021*; *Padilla et al., 2020*). We consider that the elevated levels of galectin-9 in the serum of active TB may be an indicator of the host immune response to Mtb infection, however, the magnitude of elevated galectin-9 is not sufficient to control Mtb infection and maintain latent infection. This is highly similar to other protective immune factors such as interferon-gamma, which is elevated in active TB as well (*El-Masry et al., 2007*; *Hasan et al., 2009*). On the other hand, mechanisms underlying the inhibition of mycobacterial growth induced by galectin-9 or anti-AG mAbs remain elusive. We propose they interfere with the activity of enzymes involved in AG biosynthesis and/or modify the physical properties of the cell wall, leading to disruption of AG side chain extension, thereby increasing Mtb vulnerability to host immunity. They may also function through the two-component system, that is commonly found in bacteria and allows bacteria to sense and respond to changes in the environment, such as nutrient availability or stress (*Glover et al., 2007*; *James et al., 2012*; *Majumdar et al., 2012*). Interactions between galectin-9 and AG in the cell wall may alter membrane permeability, which restrains nutrient uptake and activates sensor proteins, causing bacterial growth arrest.

Aside from their direct anti-Mtb activity, anti-AG antibodies in the serum of TB patients probably also opsonize Mtb, thereby promoting phagocytosis by mononuclear phagocytes (*Lu et al., 2019*; *Chen et al., 2016*). However, the mechanisms by which galectin-9 or antibodies inhibit mycobacterial growth depend on the details of the molecular interactions and require further investigation.

Our knowledge about antibodies which target glycans is scarce, not the least due to technical challenges. Glycan antigens have been identified on the surface of numerous microorganisms and are also expressed by certain cancer cells. Antibodies that recognize and bind these glycan antigens, therefore, are promising candidates for therapy and diagnosis of infectious and malignant diseases. For instance, the mAb 2G12 neutralizes human immunodeficiency virus-1 (HIV-1) by recognizing oligomannose-type N-glycans on the HIV-1 gp120 envelope protein, and the mAb FH6 specifically binds the Sialyl Lewis X (SLeX) antigen on the surface of various cancer cells (*Trkola et al., 1996*; *Fukushi et al., 1984*; *Kannagi et al., 1986*). In this study, we not only discovered anti-AG mAbs which directly impair Mtb growth, but also developed an efficient high-throughput screening for identifying mAbs with specificity for glycans.

In conclusion, we (i) discovered a novel cell-autonomous mechanism by which galectin-9 protects against TB via targeting AG in the cell wall of Mtb, (ii) identified neutralizing antibodies against AG in serum of TB patients, (iii) selected anti-AG mAbs for passive immunization against TB by means of a mAb screening array, (iv) characterized inhibition of Mtb replication by induction of cell wall swelling as critical mechanism of protection through AG targeting (*Figure 7E*). Our findings, thus, not only provide deeper insights into humoral immune mechanisms involved in protection against TB, but also serve as a basis for new intervention strategies against TB in adjunct to chemotherapy.

## Materials and methods

### Bacteria

Mtb H37Rv, *Mycobacterium bovis* BCG, and *Mycobacterium smegmatis* mc$^2$ 155, were from Shanghai Key Laboratory of Tuberculosis. H37Rv-GFP, and Mycobacterium bovis BCG-DsRed were generated and provided by Stefan HE Kaufman Lab. They were grown in Middlebrook 7H9 (Becton Dickinson, Cockeysville, MD) liquid medium supplemented with 0.25% glycerol, 10% oleic acid–albumin-dextrose-catalase (OADC) (Becton Dickinson, Sparks, MD) and 0.05% Tween-80.

### Cell lines

THP-1 cell line (human, acute monocytic leukemia, TIB-202) were from the American Type Culture Collection (ATCC). The identity of THP-1 has been authenticated with STR profiling by a provider. THP-1 cell line was tested negative for mycoplasma contamination.

### Clinical serum specimens

Clinical serum specimens were collected from Shanghai Pulmonary Hospital (Shanghai, PR China) from individuals who had all received BCG vaccination within 24 hr after birth. The diagnosis of TB was based on sputum culture, the presence of acid-fast bacilli in sputum smear, clinical presentation, and radiological signs. Patients who were culture-positive, sputum smear-positive, and showed signs, symptoms, or abnormal chest X-ray results were considered to have active TB. All TB patients were human immunodeficiency virus (HIV) negative and had not received anti-TB treatment. The controls were recruited from a pool of individuals who participated in a health examination program and had not been tested for antibodies to HIV. To detect the concentrations of galectin-9 in serum, 40 active TB patients and 40 healthy donors were included. For the analysis of anti-AG IgG antibody levels, 25 active TB patients and 17 healthy donors were included. All participants were between the ages of 50 and 65, of Han ethnicity, with an equal representation of males and females.

### Recombinant galectin-9 preparation

Recombinant galectin-9 protein was generated and purified as previously described (*Wu et al., 2021*). In brief, human galectin-9 cDNA was subcloned into a pET28a vector and subsequently transfected into BL21 (DE3) competent *E. coli*. Bacteria were cultivated in a Luria-Bertani (LB) liquid medium and induced overnight at 16 °C with isopropyl β-D-1-thiogalactopyranoside. Recombinant proteins were then purified from the bacterial lysates using a Ni-chelating Sepharose Fast Flow (SFF) column (GE Healthcare, Little Chalfont, UK). The concentration of galectin-9 protein was determined using a Pierce BCA Protein Assay Kit (Thermo Fisher Scientific, 23227).

### In vitro growth of mycobacteria

Mycobacteria were harvested at the mid-log phase and diluted to a calculated starting OD600 of 0.25 and added to 96-well culture plates containing Middlebrook 7H9 liquid medium together with antibody or galectin-9. OD600 of each time point of each strain was tested in real time by a Bioscreen C microplate incubator (Labsystems, USA, 23227) at 37°C.

### CFU assay

Mycobacteria were harvested at mid-log phase and diluted to a calculated starting OD600 of 0.25 and incubated in Middlebrook 7H9 liquid medium with or without antibody or galectin-9 for 30 hr at 37 °C. Appropriate dilutions were plated on 7H10 agar plates for enumeration of CFU.

### Quantification of galectin-9 by ELISA

The concentration of galectin-9 in serum was determined with the Human Galectin-9 ELISA Kit (Abcam, ab213786) according to the manufacturer's instructions.

### Immunofluorescence assay

For colocalization of galectin-9 and mycobacteria, immunofluorescence assays were performed as described previously (*Liu et al., 2018*). Briefly, THP-1 cells (ATCC, TIB-202) were infected with bacteria for 2 hr, fixed with 4% formaldehyde for 30 min at R.T., permeabilized with 0.1% Triton X-100 in PBS

for 5 min, and blocked with 5% BSA in PBS for 60 min at R.T. Cells were stained with the anti-galectin-9 antibody (Cell Signaling Technology, Cat#54330, RRID:AB_2799456), antibodies at a dilution of 1:200 in 5% BSA in PBS overnight at 4 °C and then incubated with Alexa Fluor 488 or 555 conjugated secondary antibodies (Thermo Fisher Scientific, Cat# A-11008; RRID: AB_143165; Cat# A32732, RRID:AB_2633281) at a dilution of 1:1000 for 2 hr at R.T. Nuclei were stained with DAPI.

For binding of anti-AG mAb with Mtb, H37Rv-GFP strains were harvested at mid-log phase. Subsequently, $2 \times 10^7$ H37Rv-GFP/100 µL FACS buffer was incubated with anti-AG mAb (mAb1 or mAb2, 20 µg/mL) at R.T. for 1 hr, and washed three times with PBST (PBS containing 0.05% Tween 20, pH 7.4) by centrifugation (12,000 g, 5 min). The resulting sediment was resuspended in 10 µL of ddH2O and smeared on microscopic slides.

Images were acquired using a Leica TCS SP8 confocal laser microscopy system (Leica Microsystems) at 63x magnification.

## Validation of anti-AG antibodies by ELISA

AG antigen (10 µg in 100 µL 0.1 mol/L NaHCO$_3$ buffer, pH 9.4) was added to wells of microwell plates and incubated overnight at 4 °C. After four rinses with PBST (PBS containing 0.05% Tween 20, pH 7.4), the wells were saturated with a blocking buffer. After four additional PBST rinses, serum or candidate anti-AG mAbs were added to each well and incubated for 1 hr at 37 °C. The wells were rinsed three times with PBS containing 0.05% Tween 20 and horseradish-peroxidase-labeled rabbit anti-human or anti-mouse IgG (100 µL/well; Sigma-Aldrich, Germany) was added to each well and incubated for 1 hr at 37 °C. Finally, the TMB substrate was added and the absorbance was measured with the Thermo Fisher Scientific Multiskan FC microplate photometer. Clinical serum specimens were collected from 17 healthy volunteers and 25 pulmonary TB patients before undergoing treatment at Shanghai Pulmonary Hospital (Shanghai, PR China), all of whom had received BCG vaccination within 24 hr after birth. The donors are between the ages of 50 and 65, ethnic Han, with an equal representation of males and females.

## High throughput screening of anti-AG antibody

Screening of anti-AG antibodies was based on a selection platform for Proteome Epitope Tag Antibody Library (PETAL) to pinpoint antibodies with high specificity for AG (*Wang et al., 2020*). In Brief, an antibody chip harboring 62208 mAbs was incubated with 10 µg AG in 10 mL incubation buffer (1 x PBS buffer containing 10% BSA) for 1 hr, followed by incubation with 10 µg galectin-9 protein in 10 mL incubation buffer for 1 hr. The chip was incubated with rabbit anti-galetin-9 antibody (1:5000 diluted in 10 mL incubation buffer; ab227046, Abcam, UK) followed by staining with FITC-labelled anti-rabbit IgG (1:5000 diluted in 10 mL incubation buffer; ab6717, Abcam, UK). Then the chip was scanned by GenePix 4200 A Microarray Scanner (Molecular Devices LLC) and analyzed by GenePix Pro 6.0 software.

## Chemical synthesis of AG

Mycobacterial AG containing 92 mono-saccharide units was synthesized following the reported procedure *Wu et al., 2017* and was used throughout the study.

## Proteomics analysis

### LC-MS/MS

Mycobacteria at the mid-log phase were diluted to a calculated starting OD600 of 0.25 and incubated in Middlebrook 7H9 liquid medium with or without mAb1 for 30 hr at 37°C. The bacterial pellets were collected followed by three washes with sterile saline. The bacterial pellets were resuspended in lysis buffer (8 M urea, 1% Protease Inhibitor Cocktail) and inactivated for 10 min at 100°C. The lysate was sonicated three times on ice using a high-intensity ultrasonic processor (Scientz) and centrifuged. The protein concentration of the lysate was determined with a BCA kit according to the manufacturer's instructions.

After trypsin digestion, peptides were dissolved in 0.1% formic acid and separated with nanoElute UHPLC system (Bruker, Germany) and subjected to Capillary source followed by the timsTOF Pro mass spectrometry. The resulting MS/MS data were processed using the Maxquant search engine (v1.6.6.0).

Tandem mass spectra were searched against Mtb strain ATCC 25618 83332 PR 20191210 database (3993 entries) concatenated with reverse decoy database.

### Enrichment of Gene Ontology analysis

GO annotations of proteins are divided into three broad categories: Biological Process, Cellular Component, and Molecular Function. For each category, a two-tailed Fisher's exact test was employed to test the enrichment of the differentially expressed protein against all identified proteins. The GO with a corrected p-value <0.05 was considered significant. *Enrichment of pathway analysis The* Kyoto Encyclopedia of Genes and Genomes (KEGG) database was used to identify enriched pathways by a two-tailed Fisher's exact test to test the enrichment of the differentially expressed protein against all identified proteins. The pathway with a corrected p-value <0.05 was considered significant. These pathways were classified into hierarchical categories according to the KEGG website. *Enrichment of protein domain analysis* For each category of proteins, the InterPro database (a resource that provides functional analysis of protein sequences by classifying them into families and predicting the presence of domains and important sites) was researched and a two-tailed Fisher's exact test was employed to test the enrichment of the differentially expressed protein against all identified proteins. Protein domains with a corrected p-value <0.05 were considered significant.

## Morphologic characterization of Mtb

Mtb strains were cultured in 96 U well culture plates containing Middlebrook 7H9 liquid medium with anti-AG antibody or without anti-AG antibody at 37°C, and were grown for 10–14 days until the formation of colonies. Morphologic characterization of tested strains in liquid medium were observed by 2 x magnifier. At the same time, 10 μl of culture were spread onto a glass slide. Smears on glass slides were fixed under ultraviolet light overnight. Glass slides were stained with Ziehl–Neelsen stain using a TB Stain Kit (Baso DIAGNOTICS TAIWAN, Zhuhai, China). Morphological characteristics or cell lengths of tested strains were observed using a Leica DM2500 microscope using the 100x objective.

For ultrastructural characteristics, strains at mid-log phase were collected, and analyzed by Tecnai transmission electron microscopy (TEM) with 160 kV according to the procedures of the manufacturer (GOODBIO, Wuhan, China). A $10^7$–$10^8$ bacterial suspension was used for TEM examination.

## Statistical analysis

The statistical significance of comparisons was analyzed with two-tailed unpaired Student's t-test or Mann-Whitney U test in GraphPad Prism version 8.0.1. p<0.05 was considered statistically significant. All data are shown as mean ± SD of two or more independent experiments performed in triplicate. Detailed statistical information on each experiment is provided in the respective figure legends.

## Materials availability statement

Bacterial strains and monoclonal antibodies generated in this study are available from the Lead Contact upon reasonable request.

## Acknowledgements

We thank Prof. Baoxue Ge for the helpful discussions. This work was supported by the project of National Key R&D Program of China (2021YFA1300902 to RZ, 2023YFC2307002 to HL); the National Natural Science Foundation of China (82271882 to HL, 82000009 to XW, 81470090 to LQ, 82201931 to JX,); Science and Technology Commission of Shanghai Municipality (22S11900700 to HL, 21DZ229800 to HL, 22ZR1452500 to JX); Tongji University Fundamental Research Funds for the Central Universities (22120220655 to JX); the National Natural Science Foundation for Excellent Young Scholars of China (81922030); and Shanghai ShuGuang Program (20SG19).

## Additional information

#### Competing interests

Zhaohui Wang, Mingqiao Wang, Rong Pan, Xun Meng: Employee of Abmart Inc. The other authors declare that no competing interests exist.

## Funding

| Funder | Grant reference number | Author |
| --- | --- | --- |
| National Key Research and Development Program of China | 2021YFA1300902 | Ruijuan Zheng |
| National Key Research and Development Program of China | 2023YFC2307002 | Haipeng Liu |
| National Natural Science Foundation of China | 82271882 | Haipeng Liu |
| National Natural Science Foundation of China | 82000009 | Xiangyang Wu |
| National Natural Science Foundation of China | 81470090 | Lianhua Qin |
| National Natural Science Foundation of China | 82201931 | Junfang Xu |
| Science and Technology Commission of Shanghai Municipality | 22S11900700 | Haipeng Liu |
| Science and Technology Commission of Shanghai Municipality | 21DZ229800 | Haipeng Liu |
| Science and Technology Commission of Shanghai Municipality | 22ZR1452500 | Junfang Xu |
| Tongji University | 22120220655 | Junfang Xu |
| Excellent Young Scientists Fund | 81922030 | Haipeng Liu |
| Shanghai Shuguang Program | 20SG19 | Haipeng Liu |

The funders had no role in study design, data collection and interpretation, or the decision to submit the work for publication.

## Author contributions

Lianhua Qin, Data curation, Methodology; Junfang Xu, Validation, Writing – original draft; Jianxia Chen, Data curation, Formal analysis; Sen Wang, Ruijuan Zheng, Xun Meng, Lu Zhang, Wei Sha, Resources, Methodology; Zhenling Cui, Rong Pan, Investigation; Zhonghua Liu, Visualization; Xiangyang Wu, Zhaohui Wang, Mingqiao Wang, Software; Jie Wang, Xiaochen Huang, Methodology; Stefan HE Kaufmann, Supervision, Writing – review and editing; Haipeng Liu, Conceptualization, Supervision, Project administration, Writing – review and editing

## Author ORCIDs

Junfang Xu ⓘ https://orcid.org/0000-0002-4150-7305
Stefan HE Kaufmann ⓘ https://orcid.org/0000-0001-9866-8268
Haipeng Liu ⓘ https://orcid.org/0000-0002-3338-6291

Reviewer #1 (Public review): https://doi.org/10.7554/eLife.92737.3.sa1
Reviewer #2 (Public review): https://doi.org/10.7554/eLife.92737.3.sa2
Author response https://doi.org/10.7554/eLife.92737.3.sa3

## Additional files

### Supplementary files
• MDAR checklist

### Data availability

The authors confirm that the data supporting the findings of this study are included within the article. The raw data of the mass spectrometry proteomics have been deposited in the ProteomeXchange Consortium via the PRIDE partner repository and are publicly accessible with the accession number: PXD056927.

The following dataset was generated:

| Author(s) | Year | Dataset title | Dataset URL | Database and Identifier |
|---|---|---|---|---|
| Liu H | 2024 | Cell-autonomous targeting of arabinogalactan by host immune factors inhibits mycobacterial growth | http://www.ebi.ac.uk/pride/archive/projects/PXD056927 | PRIDE, PXD056927 |

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
