## [Editor Report · eLife Assessment]

The main idea tested in this work is that host galectin-9 inhibits Mycobacterium tuberculosis (Mtb) growth by recognizing the Mtb cell wall component arabinogalactan (AG) and, as a result, disrupting mycobacterial cell wall structure. Moreover, a similar effect is achieved by anti-AG antibodies. While the hypothesis is intriguing and the work has the potential to make a **valuable** contribution to Mtb therapy, the evidence presented is **incomplete** and does not explain several critical points including the dose-independent effect of galectin-9 on Mtb growth and how anti-AG antibodies and galectin-9 access the AG layer of intact Mtb.

---

## [Referee Report · Reviewer #1 (Public review)]

The molecular interactions which determine infection (and disease) trajectory following human exposure to Mycobacterium tuberculosis (Mtb) are critical to understanding mycobacterial pathogenicity and tuberculosis (TB), a global public health threat which disproportionately impacts a number of high-burden countries and, owing to the emergence of multidrug-resistant Mtb strains, is a major contributor to antimicrobial resistance (AMR). In this submission, Qin and colleagues extend their own previous work which identified a potential role for host galectin-9 in recognizing the major Mtb cell wall component, arabinogalactan (AG). First, the authors present data indicating that galectin-9 inhibits mycobacterial growth during in vitro culture in liquid and on solid media, and that the inhibition depends on carbohydrate recognition by galectin-9. Next, the authors identify anti-AG antibodies in sera of TB patients and use this observation to inform isolation of monoclonal anti-AG antibodies (mAbs) via an in vitro screen. Finally, they apply the identified anti-AG mAbs to inhibit Mtb growth in vitro via a mechanism which proteomic and microscopic analyses suggest is dependent on disruption of cell wall structure. In summary, the dual observation of (i) the apparent role of naturally arising host anti-AG antibodies to control infection and (ii) the potential utility of anti-AG monoclonal antibodies as novel anti-Mtb therapeutics is compelling; however, as noted in the comments below, the evidence presented to support these insights is not adequate and the authors should address the following:

(1) The experiment which utilizes lactose or glucose supplementation to infer the importance of carbohydrate recognition by galectin-9 cannot be interpreted unequivocally owing to the growth-enhancing effect of lactose supplementation on Mtb during liquid culture in vitro.

(2) Similar to the comment above, the apparent dose-independent effect of galectin-9 on Mtb growth in vitro is difficult to reconcile with the interpretation that galectin is functioning as claimed.

(3) The claimed differences in galectin-9 concentration in sera from tuberculin skin test (TST)-negative or TST-positive non-TB cases versus active TB patients are not immediately apparent from the data presented.

(4) Neither fluorescence microscopy nor electron microscopy analyses are supported by high-quality, interpretable images which, in the absence of supporting quantitative data, renders any claims of anti-AG mAb specificity (fluorescence microscopy) or putative mAb-mediated cell wall swelling (electron microscopy) highly speculative.

(5) Finally, the absence of any discussion of how anti-AG antibodies (similarly, galectin-9) gain access to the AG layer in the outer membrane of intact Mtb bacilli (which may additionally possess an extracellular capsule/coat) is a critical omission - situating these results in the context of current knowledge about Mtb cellular structure (especially the mycobacterial outer membrane) is essential for plausibility of the inferred galectin-9 and anti-AG mAb activities.

---

## [Referee Report · Reviewer #2 (Public review)]

Summary:

In this manuscript, the authors work to extend their previous observation that galectin-9 interacts with arabinogalactans of Mtb in their EMBO reports 2021 manuscript. Here they provide evidence for the CARD2 domain of galectin-9 can inhibit the growth of Mtb in culture. In addition, antibodies that also bind to AG appear to inhibit Mtb growth in culture. These data indicate that independent of the common cell-associated responses to galectin-9 and antibodies, interaction of these proteins with AG of mycobacteria may have consequences for bacterial growth.

Strengths:

The authors provided several lines of evidence in culture media that the introduction of galectin-9 proteins and antibodies inhibit the growth rate of Mtb.

Weaknesses:

The methodology for generating and screening the anti-AG antibodies lacks pertinent details for recapitulating and interpreting the results.

The figure legends and methods associated with the microscopy assays lack sufficient details to appropriately interpret the experiments conducted.

The galectin-9 measured in the sera of TB patients does not approach the concentrations required for Mtb growth restriction in the in vitro assays performed by the authors. It remains difficult to envision how greater levels of galectin-9 release might contribute to Mtb control in severe forms of TB, since higher levels of serum Gal9 has been observed in other human studies and correlate with poorly controlled infection. The authors over-interpret the role of Gal9 in bacterial control during disease/infection without any evidence of impact on in vivo (animal model) control.

---

## [Author Response]

The following is the authors’ response to the original reviews.

**Reviewer #1 (Public Review):**
Question 1: The experiment that utilizes lactose or glucose supplementation to infer the importance of carbohydrate recognition by galectin-9 cannot be interpreted unequivocally owing to the growth-enhancing effect of lactose supplementation on Mtb during liquid culture in vitro.

Thank you for this very constructive comment. We repeated the experiments by lowering the concentration of lactose or AG from 10 μg/mL to 1 μg/mL. We found that low concentration of lactose or AG showed neglectable effect on Mtb growth, however, they still reversed the inhibitory effect of galectin-9 on mycobacterial growth (revised Fig. 2A, C). Therefore, we consider that the supplementation of lactose or AG reverse galectin-9 mediated inhibition of Mtb growth largely through its carbohydrate recognition rather than their growth-enhancing effect.

Question 2: Similar to the comment above, the apparent dose-independent effect of galectin-9 on Mtb growth in vitro is difficult to reconcile with the interpretation that galectin is functioning as claimed.

We thank the reviewer for the correction. Indeed, as the reviewer pointed out, galectin-9 inhibits Mtb growth in dose-independent manner. We had corrected the claim in the revised manuscript (Line 114).

Question 3: The claimed differences in galectin-9 concentration in sera from tuberculin skin test (TST)-negative or TST-positive non-TB cases versus active TB patients are not immediately apparent from the data presented.

We appreciate your concern. Previous samples are from a cohort set up in Max Plank Institute for Infection Biology. We have performed the detection of galectin-9 in sera in another independent cohort of active TB patients and healthy donors in China. And we found higher abundance of galectin-9 in serum from TB patients than that from heathy donors (revised Fig. 1E).

Question 4: Neither fluorescence microscopy nor electron microscopy analyses are supported by high-quality, interpretable images which, in the absence of supporting quantitative data, renders any claims of anti-AG mAb specificity (fluorescence microscopy) or putative mAb-mediated cell wall swelling (electron microscopy) highly speculative.

We appreciate your concern. We have improved the procedure of the immunofluorescence assay and obtained high-quality and interpretable images with quantitative data (revised Fig. 4F). As for electron microscopy analyses, we added clearer label indicating cell wall in revised manuscript (revised Fig. 7C).

Question 5: Finally, the absence of any discussion of how anti-AG antibodies (similarly, galectin-9) gain access to the AG layer in the outer membrane of intact Mtb bacilli (which may additionally possess an extracellular capsule/coat) is a critical omission - situating these results in the context of current knowledge about Mtb cellular structure (especially the mycobacterial outer membrane) is essential for plausibility of the inferred galectin-9 and anti-AG mAb activities.

Exactly, AG is hidden by mycolic acids in the outer layer of Mtb cell wall. As we have discussed in the Discussion part of previous manuscript (line 285), we speculate that during Mtb replication, cell wall synthesis is active and AG becomes exposed, thereby facilitating its binding to galectin-9 or AG antibody and leading to Mtb growth arrest. It’s highly possible that galectin-9 or AG antibody targets replicating Mtb.

**To Reviewer #2 (Public Review):**
Question 1: In light of other observations that cleaved galectin-9 levels in the plasma is a biomarker for severe infection (Padilla A et al Biomolecules 2021 and Iwasaki-Hozumi H et al. Biomoleucles 2021) it is difficult to reconcile the author's interpretation that the elevated gal-9 in Active TB patients (Figure 1E) contributes to the maintenance of latent infection in humans. The authors should consider incorporating these observations in the interpretation of their own results.

Thank you for these very insightful comments. We observed elevated levels of galectin-9 in the serum of active TB patients, consistent with reports indicating that cleaved galectin-9 levels in the serum serve as a biomarker for severe infection (Iwasaki-Hozumi et al., 2021; Padilla et al., 2020). We consider that the elevated levels of galectin-9 in the serum of active TB may be an indicator of the host immune response to Mtb infection, however, the magnitude of elevated galectin-9 is not sufficient to control Mtb infection and maintain latent infection. This is highly similar to other protective immune factors such as interferon gamma, which is elevated in active TB as well (El-Masry et al., 2007; Hasan et al., 2009). We have included the discussion in the revised manuscript (line 298).

Question 2: The anti-AG titers were measured only in individuals with active TB (Figure 3C), generally thought to be a less protective immunological state. The speculation that individuals with anti-AG titers have some protection is not founded. Further only 2 mAbs were tested to demonstrate restriction of Mtb in culture. It is possible that clones of different affinities for AG present within a patient's polyclonal AG-antibody responses may or may not display a direct growth restriction pressure on Mtb in culture. The authors should soften the claims about the presence of AG-titers in TB patients being indicative of protection.

We appreciate your concern. As per your suggestion, we have softened the claim to that “We speculate that during Mtb infection, anti-AG IgG antibodies are induced, which potentially contribute to protection against TB by directly inhibiting Mtb replication albeit seemingly in vain.”

References

El-Masry, S., Lotfy, M., Nasif, W.A., El-Kady, I.M., and Al-Badrawy, M. (2007). Elevated serum level of interleukin (IL)-18, interferon (IFN)-gamma and soluble Fas in patients with pulmonary complications in tuberculosis. Acta microbiologica et immunologica Hungarica *54*, 65-77.

Hasan, Z., Jamil, B., Khan, J., Ali, R., Khan, M.A., Nasir, N., Yusuf, M.S., Jamil, S., Irfan, M., and Hussain, R. (2009). Relationship between circulating levels of IFN-gamma, IL-10, CXCL9 and CCL2 in pulmonary and extrapulmonary tuberculosis is dependent on disease severity. Scandinavian journal of immunology *69*, 259-267.

Iwasaki-Hozumi, H., Chagan-Yasutan, H., Ashino, Y., and Hattori, T. (2021). Blood Levels of Galectin-9, an Immuno-Regulating Molecule, Reflect the Severity for the Acute and Chronic Infectious Diseases. Biomolecules *11*.

Padilla, S.T., Niki, T., Furushima, D., Bai, G., Chagan-Yasutan, H., Telan, E.F., Tactacan-Abrenica, R.J., Maeda, Y., Solante, R., and Hattori, T. (2020). Plasma Levels of a Cleaved Form of Galectin-9 Are the Most Sensitive Biomarkers of Acquired Immune Deficiency Syndrome and Tuberculosis Coinfection. Biomolecules *10*.